# Hepatitis B Virus (HBV) Reactivation Following Pharmacological Eradication of Hepatitis C Virus (HCV)

**DOI:** 10.3390/v11090850

**Published:** 2019-09-13

**Authors:** Mariantonietta Pisaturo, Margherita Macera, Loredana Alessio, Federica Calò, Nicola Coppola

**Affiliations:** 1Laboratory for the identification of prognostic factors of response to the treatment against infectious diseases, University of Campania, 80138 Naples, Italy; meri.pisaturo@libero.it; 2Infectious Diseases and Viral Hepatitis, Department of Mental and Physical Health and Preventive Medicine, University of Campania, 80138 Naples, Italy; macera.margherita@libero.it (M.M.); fede.calo85@gmail.com (F.C.); 3Infectious Diseases Unit, AO Caserta, 81100 Caserta, Italy; loredana.alessio@gmail.com

**Keywords:** HBV/HCV dual infection, HBsAg, occult HBV infection, overt HBV reactivation, occult HBV reactivation

## Abstract

The US Food and Drug Administration issued a black box warning related to the risk of reactivation of overt/occult hepatitis B virus (HBV) infection during direct acting-antivirals (DAA) treatment. This review evaluated the prevalence of HBV reactivation after hepatitis C virus (HCV) pharmacological suppression and hypothesized the management and prevention of this reactivation. During and after DAA-based treatment, reactivation of HBV infection is common in patients with detectable serum HBsAg (from 2% to 57%) and very low (less than 3%) in individuals with isolated anti-HBc antibodies. The severity of hepatic damage may range from HBV reactivation without hepatitis to fulminant hepatic failure requiring liver transplantation. Thus, HBsAg-positive patients should receive nucleo(s)tide analog (NA) treatment or prophylaxis at the same time as DAA therapy. For those patients with occult B infection, there are no sufficient recommendations to start prophylactic treatment. Reactivation of overt or occult HBV infection during or after eradication of HCV infection is an issue to consider, and additional studies would help to determine the best management of this virological and clinical event.

## 1. Introduction

Hepatitis B virus (HBV) and hepatitis C virus (HCV) infections are among the leading causes of chronic liver disease worldwide, including liver cirrhosis and hepatocellular carcinoma (HCC) [1,2].

Due to the shared modes of transmission, HBV/HCV dual infection is not uncommon in highly endemic areas and among subjects with a high risk of parenteral transmission, with an incidence reported to be in the range of 1%–15% worldwide [3,4]. Due to a lack of large-scale population-based studies, the exact number of HBV/HCV infected patients is unknown. The reported prevalence of HBV/HCV dual infection in different studies reveals wide differences depending on the geographical region, study population, method of patient-selection criteria and study design. Clinical studies have shown that 2%–10% of patients with chronic HCV infection circulate HBsAg [5,6,7], and that 5%–20% of patients with chronic hepatitis B are anti-HCV-positive [8,9].

HBV/HCV dual infection was characterized by a reciprocal inhibition of viral genomes [10,11,12,13,14], a dynamic fluctuation of HBV and HCV viremia [15], and a spontaneous clearance over time of one of the two viruses in longitudinal investigations [16,17,18]. 

The reciprocal inhibition may also be influenced by antiviral therapy. In the Interferon era, the pharmacological inhibition of one virus was associated with the reactivation of the other in about 30% of cases [19,20,21]. 

Recently, the use of direct-acting antivirals (DAAs) has revolutionized the care of HCV-infected patients with a very high rate of sustained virological response. However, also in the DAA era, the reactivation of HBV in patients treated for HCV was observed, both in overt and occult HBV infection (negative hepatitis B surface antigen, but detectable liver and/or serum HBV DNA). In fact, in October 2016, the US Food and Drug Administration issued a black box warning related to the risk of reactivation of overt/occult HBV infection in persons treated with DAAs [22], and a growing body of evidence supports this hypothesis [23,24,25]. 

However, since the data available in the literature on this topic are fragmented, this review evaluated the prevalence of HBV reactivation after HCV pharmacological suppression and hypothesized the management and prevention of this reactivation. 

## 2. Virological and Clinical Characteristics of HBV/HCV Dual Infection

### 2.1. Virological Interaction

The virological and molecular aspects of HBV/HCV dual infection are only partially understood.

Some in-vitro studies suggested a non-interference between HBV and HCV. In fact, liver cells with active HBV replication may be infected also by HCV [9] and HBV, and HCV can replicate in the same hepatocytes [26,27,28,29]. However, other in-vitro studies provided data in favor of a reciprocal suppression or of viral interference [30,31] and demonstrated that the HCV “core” protein strongly inhibits HBV replication [32,33]. It has also been shown that the HCV NS5A protein may influence HBV activity [34,35], but these contrasting data do not allow any conclusions to be drawn on this point.

Most cross-sectional studies evaluated the viral load of both viruses at a single check point and reported a strong inhibitory effect exerted by the super-infecting virus on the pre-existing virus [10,11,12,15,17]. In a single one-year longitudinal study, the virological profile of chronic HBV/HCV coinfection was characterized by dynamic fluctuations in HBV and HCV viremia in one-third of cases, whereas in the remaining two-thirds it remained stable [16]. A spontaneous clearance of both viruses has been observed in two longitudinal studies [16,17]. In particular, in a six-year follow-up study of untreated patients, Sheen et al. found a rate of HBsAg clearance 2.5 times higher in HBsAg/anti-HCV-positive patients with chronic hepatitis than in those with HBV chronic infection alone [16].

Another interesting clinical presentation is the role in HBsAg-negative subjects of occult HBV infection, i.e., the presence of HBV DNA in liver tissue in anti-HCV-positive patients. Occult HBV infection has been identified in up to 50% of patients with chronic HCV infection [36,37], probably as a conclusive event of the virological reciprocal inhibition.

### 2.2. Clinical Presentation

Patients with dual HBV/HCV infection show a large spectrum of clinical profiles. Acute HBV/HCV dual infection is rare and mostly confined to special populations with a higher risk of acquiring both infections, such as intravenous drug users, men having sex with men, and poly-transfused subjects [38,39]. One or both viruses may be cleared after the acute episode [40], but in a substantial proportion of cases, the disease evolves to HBV, HCV or HBV/HCV chronic infection, and in a minority of cases, to fulminant hepatitis. The chronicity rates were comparable to patients with monoinfection with either of the viruses [15,41,42].

HCV superinfection in HBsAg chronic carriers is frequent in endemic areas of HBV infection, such as Asia, South America and sub-Saharan Africa [43]. In HBeAg-positive patients, this event may result in seroconversion to anti-HBe and in some cases to HBsAg clearance [11]. Fulminant hepatic failure was significantly higher among patients with underlying HBV infection than those without (23% vs. 3%) [44]. It has also been reported that in HBsAg chronic carriers, HCV superinfection may lead to liver cirrhosis and HCC more frequently than hepatitis delta virus (HDV) superinfection [45].

As regards HBV superinfection in chronic HCV carriers, Sagnelli et al. [11] showed that HBV superinfection may be associated with acute deterioration of liver function with an increased risk of fulminant hepatitis. Subsequently, the same authors [46] demonstrated in a long-term follow-up study a sustained clearance of HCV infection in about 1/3 of 29 chronic HCV-RNA-positive patients after acute hepatitis B.

In most cases HBV/HCV dual infection is diagnosed in the chronic phase.

The data on the degree of liver damage between coinfected and monoinfected patients are conflicting. No difference was reported by some authors [4,47], but other authors suggested far more severe necroinflammation and fibrosis in patients with coinfection [48,49,50]. Moreover, a more frequent progression to liver cirrhosis, hepatic decompensation or HCC was reported for HBV/HCV dual infection than HBV or HCV monoinfections [38,51,52].

As regards the clinical impact of occult HBV infection in patients with chronic hepatitis C, the data seem to suggest that it was associated with more severe liver damage and with the development of HCC [36,53].

## 3. HBV Reactivation Due to Pharmacological Suppression

The interaction between HBV and HCV may also be evident during the pharmacological suppression of one of the two viruses with a possible virological and clinical reactivation of the other virus. The virological reactivation of overt or occult HBV infection is defined as a de novo detection of HBV DNA in individuals with no previously detectable HBV DNA or a 1 to 2 log IU/mL rise in serum HBV DNA levels or hepatitis B surface antigen (HBsAg) sero-reversion in the HBsAg-negative individuals. Clinical reactivation is defined as an increase in alanine-aminotransferase serum values in the presence of HCV virological reactivation [25,54].

There are several models of HBV/HCV interaction due to pharmacological suppression considering the type of antiviral treatment and the inhibited virus. These models are described below.

### 3.1. Overt or Occult HBV Reactivation during and after IFN-Based HCV Therapy

Before the introduction of DAA therapy for HCV, the treatment of hepatitis C in dual hepatitis B/C patients with active hepatitis C was Pegylated Interferon (Peg-INF) plus Ribavirin (RBV) [55].

Table 1 describes the different studies analyzing the data of HBV reactivation during therapy with Peg-INF plus RBV. Potthoff et al. described HBV reactivation in 4 (31%) of 13 HBV-DNA-negative patients after anti-HCV treatment with Peg-IFN and RBV [19]. In a study by Liu et al. [20], 36.4% of 77 HBV/HCV patients with undetectable serum HBV DNA at baseline became positive under Peg-IFN plus RBV treatment used to eradicate HCV infection; in these patients viral HBV reactivation was not associated with clinical events, apart from an increase in some patients in serum ALT levels never exceeding 5-fold the upper normal value. However, evaluated for an additional four-year follow-up period, HBV DNA became positive in a total of 47 (61.8%) of 77 cases enrolled, with a transient reappearance in 21 (44.7%), intermittent in 12 (25.5%) and sustained in 14 (29.8%). In full agreement with these data, Yu et al. [21] observed that 11 (23.9%) of 46 patients with HBV/HCV dual infection (all HBV-DNA-negative at baseline) became HBV-DNA-positive after Peg-IFN plus RBV treatment and that HBV reactivation was more frequent in patients who achieved SVR (33.3%) than in non-responders (8.7%).

Few studies have described the consequence of pharmacological suppression in HCV-positive patients with occult HBV infection treated with peg-IFN plus RBV. In 160 HBsAg-negative/HCV-infected patients treated, Liu et al. [20] described the appearance of HBV DNA in 10 (6.3%) (positivity for serum hepatitis B core antibody in six and for serum antibody to HBsAg in eight); the median serum HBV-DNA level of the 10 patients with virological reactivation was 1831 IU/mL (range, 478–12,000 IU/mL). However, all 10 HCV monoinfected patients became HBV-DNA undetectable at the end of therapy and none showed a clinical HBV reactivation. Similarly, in a study by Szymanek-Pasternak et al. [56] of the 99 HCV/anti-HBc-positive patients followed-up for 24 weeks after the end of treatment with Peg-IFNα and RBV, none developed HBV reactivation. 

Moreover, occult HBV infection did not influence the efficacy of anti-HCV treatment [57,58]. For example, Chen et al. [57] in a retrospective study on 126 consecutive chronic hepatitis C patients demonstrated that only six were HBV-DNA-positive before antiviral treatment, with no impact on the efficacy of peg-IFN plus RBV. Summing up, according to the results of the studies reported in the text, interferon-based therapeutic regimens had no impact on the outcome of both overt and occult HBV infection.

### 3.2. Overt or Occult HBV Reactivation during and after DAA Therapy

Reactivation of overt or occult HBV infection has emerged as an important clinical event in patients treated for hepatitis C virus with direct-acting antivirals (DAAs).

Table 2 shows the studies analyzing HBV reactivation associated with DAA therapy. However, we underline that the definition of HBV reactivation varied in the different studies.

The FDA identified 29 reports of HBV reactivation in patients receiving DAAs from 22 November 2013 to 15 October 2016. Of these initial cases, two resulted in death, while one required liver transplantation [59].

In subsequent cohort studies, HBV reactivation was frequent in patients with detectable serum HBsAg and rare in individuals with isolated anti-HBc antibodies; the severity of hepatic damage ranges from HBV reactivation without hepatitis to fulminant hepatic failure requiring liver transplantation. 

Wang et al. [24] evaluated 327 Chinese patients receiving DAAs; 10 were HBsAg-positive and 124 patients had occult HBV infection, and HBV reactivation was identified in three HBsAg-positive patients (one was asymptomatic, one with jaundice and one with liver failure).

In Taiwan, HBV reactivation was not observed among 57 patients with past HBV infection, but was found in 4 of 7 patients with a current infection; clinical reactivation of HBV was observed in one patient with pre-treatment detectable HBV DNA who recovered after entecavir administration, and the other three patients with HBV virological reactivation showed a reappearance of low-level HBV DNA without a clinical reactivation [60].

Virological HBV reactivation can occur both during and after DAA therapy. In fact, the potent DAAs suppressing HCV replication may lead to increased HBV replication more quickly than previous HCV regimens. For example, in the study by Macera et al. [25], 5 (17%) of the 29 HBsAg-positive patients treated with DAAs showed HBV reactivation, in particular, two at month 1 of DAA treatment, one at month 3, one at month 4 and one at the end of DAA treatment.

Among 848 individuals treated with DAAs, no HBV reactivation was observed in HBsAg-negative/anti-HBc-positive patients. In contrast, 5 of 9 HBsAg-positive patients experienced HBV reactivation, and 3 of them required HBV treatment [61]. 

In conclusion, the studies described in Table 2 suggested that the risk of HBV reactivation is frequent in patients with overt infection but negligible in those with occult infection [62,63,64,65].

In agreement with these data, in a meta-analysis comparing reactivation rates in patients with chronic versus occult infection hepatitis B virus, reactivation was less common among individuals with occult HBV infection. [67]. 

In a recent meta-analysis, the overall risk of virological HBV reactivation in patients with untreated chronic and resolved HBV was 24% and 1.4%, respectively [68]. The risk of HBV- reactivation-related hepatitis was 9% in patients with chronic HBV infection, whereas no HBV-related hepatitis was reported in patients with a resolved infection. However, two other important concepts emerged from this meta-analysis: the risk of virological HBV reactivation was comparable between patients with HBV DNA (<20 UI/mL) versus patients with quantifiable HBV DNA (≥20 IU/mL) at baseline; and the risk of virological HBV reactivation and clinical HBV was not significantly different in patients with cirrhosis versus patients without cirrhosis [68]. 

Finally, it is necessary to test for HBV markers before starting DAAs, and HBsAg-positive patients should be considered for concomitant NA prophylaxis, while the HBsAg-negative/anti-HBc-positive patients should be monitored and tested for HBV reactivation in the case of ALT elevation.

### 3.3. HCV Reactivation during and after HBV Therapy

When HBV replication is inhibited by antiviral therapy, a reactivation of HCV infection is possible. 

The treatment strategies of chronic hepatitis B include a high genetic barrier nucleos(t)ide analogue (NUC), such as entecavir or tenofovir, and/or interferon alpha-based immunomodulation, such as Peg-IFN [18,52].

Scant information is available on HCV reactivation in HBV/HCV patients treated with NUC or Peg-IFN. HCV reactivation, defined as an increase in viral load to at least one log10 IU/mL above the baseline value, is relatively infrequent in NUC treatment for HBV suppression, usually without clinical events, and associated, only in a minority of cases, with an abnormal enzymatic profile.

PegIFN-based therapy has some antiviral activity against both HBV and HCV. It is effective for HBV infection in about 30% of cases and for HCV infection when combined with ribavirin in about 50%–60% [69]. For this reason, HCV reactivation during HBV therapy with an IFN-based regimen is a very rare event.

Coppola et al. [70], in a study on the tolerability and efficacy of NUC treatment in a cohort of 24 HBV/HCV HBV-DNA-positive cirrhotic patients treated for 18 months, described an HBV DNA clearance in 96% of patients and an HCV reactivation as a relatively rare event, occurring only in 3 (12.5%) patients, but not associated with any clinical deterioration.

Overall, since HCV is typically dominant, HCV reactivation is rarely observed in the case of pharmacological suppression of HBV infection. However, monitoring of HCV replication remains mandatory for a correct diagnosis and a proper therapeutic approach.

## 4. Management of HBV Reactivation during or after DAA Therapy

HBV reactivation as a consequence of anti-HCV treatment is an event to consider because, as mentioned above, when HCV is pharmacologically suppressed its inhibitory effects on HBV replication can be released, resulting in possible HBV reactivation [18]. Interferon-based anti-HCV therapies have rarely led to HBV reactivation due to their suppressive effect on both HBV and HCV replication [21]. The introduction of DAA treatment has increased the risk of HBV reactivation. In fact, in the literature, various studies have described HBV reactivation after a successful clearance of hepatitis C through the use of DAA therapy in patients with HCV/HBV coinfection [24,25,55,71,72].

This clinical event is common in patients with detectable serum HBsAg and lower in individuals with isolated anti-HBc antibodies, and the severity of hepatic damage may range from HBV reactivation without hepatitis to fulminant hepatic failure requiring liver transplantation. In fact, considering HBsAg-positive patients treated with DAAs, the prevalence of HBV reactivation is from 2% to 57%, while the HBsAg-negative/anti-HBc-positive is from 0% to 3%.

Figure 1 shows how, in our opinion, HBV infection should be managed during and after the eradication of HCV infection. Also in agreement with the guidelines of the European Association for the Study of the Liver, before the start of DAA all patients should be screened for HBsAg and anti-HBs and anti-HBc. The HBsAg-positive patients who fulfill the standard criteria for HBV treatment should receive nucleo(s)tide analog (NA) treatment at the same time as DAA therapy. Those who do not fulfill the criteria should be treated prophylactically with NA during, and 12 weeks following, DAA treatment, except for patients with liver cirrhosis for whom NA should be continued [50,55]. For those patients with occult hepatitis B infection there are no sufficient recommendations to start prophylactic treatment. The HBsAg-negative patients at risk of occult HBV infection (anti-HBc-positive subjects) should be monitored for HBsAg every three months during DAA and for 12 weeks after stopping treatment.

If HBV reactivation occurs during or after DAA therapy, NA treatment should be initiated as soon as possible.

## 5. Conclusions

Reactivation of overt or occult HBV infection during or after eradication of HCV infection is an issue to consider, and additional studies would help to determine the best management of this virological and clinical event.

It would be better to define, with clinical studies, which patients actually need treatment or prophylaxis in the case of an overt HBV/HCV dual infection, and how and how often to monitor patients with occult HBV/HCV dual infection. However, with appropriate screening, monitoring, and treatment, HBV reactivation is a manageable event, so DAAs remain a safe and highly effective treatment for HCV infection.

## Figures and Tables

**Figure 1 viruses-11-00850-f001:**
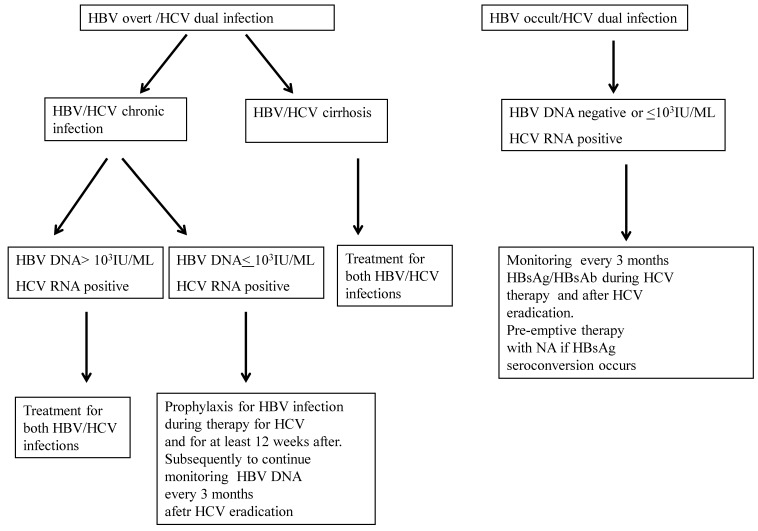
Management of HBV infection during and after DAA-based treatment. Footnotes: NA, nucleo(s)tide analog.

**Table 1 viruses-11-00850-t001:** Studies evaluating hepatitis B virus (HBV) reactivation in anti-hepatitis C virus (HCV)-positive patients treated with Interferon-based therapy.

Author, References	Study	Country	Anti HCV Therapy	Sample Size	Definition of Overt Reactivation	Reactivation Overt HBV Infection N/%	Definition of Occult Reactivation	Reactivation Occult HBV Infection N/%
Potthoff et al. [19].	Prospective multicenter study	Germany	Peg-IFN plus rbv 48 weeks	6 active HBV 13 inactive HBV	Increase of HBV-DNA >2000 UI/mL	4 (31%) inactive HBV patients with undetectable HBV DNA became HBV-DNA detectable	//	//
Liu et al. [20].	Prospective multicenter study	Taiwan	Peg-IFN plus rbv	68 HBV-DNA >200 UI/Ml 77 HBV-DNA <200 UI/mL	Increase HBV-DNA >=200 UI/mL	28 (36.4%) HBV-DNA <200 UI/mL Reactivation	//	//
Yu et al. [21].	Clinical Trials	Taiwan	Peg-IFN plus rbv	62 active HBV 77 patients inactive HBV	Increase HBV-DNA >=200 UI/mL	47 (61.8%) patients with inactive HBV developing hepatitis B reappearance	//	//
Szymanek-Pasternak [56].	Retrospective study	Poland	Peg-IFN plus rbv	99 occult HBV	Not determined	//	No data	0 (0%) reactivation HBV

**Table 2 viruses-11-00850-t002:** Studies evaluating HBV reactivation in anti-HCV-positive patients treated with DAA-based therapy.

Author, References	Study	Country	Sample Size N° Overt HBV Infection/N° HBV Occult Infection	Definition of Reactivation	Reactivation Overt HBV Infection N/%	Reactivation Occult HBV Infection N/%	Anti-HCV Therapy
Wang et al. [24]	Observational study	China	10/124	For overt infection, “Hepatitis” was defined as a more than 2-fold increase of ALT on two consecutive determinations at least five days apart, from the nadir during DAA therapy and follow-up. For occult HBV infection, reactivation of past HBV infection was defined as one of the following: (1) HBsAg turning from negative to positive (2) Appearance of HBV DNA in absence of HBsAg (3) HBV DNA turned from undetectable to detectable in HBsAg-negative patients.	3(30%)	0	Multiple IFN-free regimens
Yeh et al. [60]	Observational study	Taiwan	7/57	For overt HBV infection: at least 1 log10 IU/mL increase from baseline for those with baseline detectable HBV DNA or a reappearance of HBV DNA for those with baseline undetectable HBV DNA. For occult HBV infection: reappearance of HBsAg, or an HBV DNA > 2000 IU/mL.	4(57%)	0	Multiple IFN-free regimens
Mucke et al. [61]	Cohort study	Europe	9/263	Increase or reappearance of HBV DNA	5(55%)	0	Multiple IFN-free regimens PLUS 1 IFN based regimen
Londono et al. [62].	Cohort study	Spain	10/64	>1 log increase in HBV-DNA levels	5(50%)	1(1.6%)	Multiple IFN-free regimens
Belperio et al. [63].	Cohort study	United States	377/22479	>1000 IU/mL increase in HBV DNA or HBsAg detection in a person who was previously negative	8(2.1%)	1(0.004%)	Multiple IFN-free regimens
Doi et al. [64].	Cohort study	Japan	4/155	Reappearance of serum HBV DNA ≥20 IU/mL following baseline undetectable HBV DNA or detectable but <20 IU/mL, or a ≥10-fold increase in HBV DNA compared with baseline	2(50%)	3(2%)	Ledipasvir/sofosbuvir or sofosbuvir plus ribavirin
Liu et al. [65].	Observational study	Taiwan	12/81	Reappearance of serum HBV DNA ≥100 IU/mL in patients with baseline undetectable viral load, or ≥2 log10 IU/mL increase in patients with baseline detectable viral load	2	0	Multiple IFN-free regimens
Macera et al. [25].	Observational study	Italy	29/0	Increase in ALT serum values of at least 2-fold the baseline values and as the development of signs or symptoms of hepatic decompensation, associated to a virological HBV reactivation	5(17%)/29	0	Multiple IFN-free regimens
Sulkowski et al. [66].	Cohort study	Taiwan and Korea	0/103	Reappearance or rise of HBV DNA in the serum of a patient with previously inactive or resolved HBV infection	0	0	Sofosbuvir + ledipasvir

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
