# Peer review of "Hepatitis B Virus (HBV) Reactivation Following Pharmacological Eradication of Hepatitis C Virus (HCV)"

_viruses, 2019, doi:10.3390/v11090850_

Round 1

Reviewer 1 Report

The review by Pisaturo et al addresses the impact of successful clearance of HCV infection on HBV reactivation. The topic is very interesting and a review of this area is timely. While the review is well laid out and references are appropriate, the language is unnecessarily complicated at times and the review would benefit from editing to improve the use of English. For example, in section 2.1, lines 64-65, the language used is excessively complicated and, combined with grammatical errors, is difficult to understand.

It would also be beneficial to provide more interpretation or discussion of the literature, as in some sections the results of studies are simply listed with no discussion of the implications of these results.

Please define the acronyms ‘NA’ and ‘NUC’ in the Figure legend for Figure 1.

Author Response

Dear Editor,

We re-submit to your attention our paper Hepatitis B Virus (HBV) Reactivation Following Pharmacological Eradication of Hepatitis C Virus (HCV) ( Viruses-586419 ) modified according to the suggestions of the reviewers.

ANSWER TO THE COMMENTS OF REVIEWER 1

Reviewer: 1

Comments and Suggestions for Authors

The review by Pisaturo et al addresses the impact of successful clearance of HCV infection on HBV reactivation. The topic is very interesting and a review of this area is timely. While the review is well laid out and references are appropriate, the language is unnecessarily complicated at times and the review would benefit from editing to improve the use of English. For example, in section 2.1, lines 64-65, the language used is excessively complicated and, combined with grammatical errors, is difficult to understand.

It would also be beneficial to provide more interpretation or discussion of the literature, as in some sections the results of studies are simply listed with no discussion of the implications of these results.

Please define the acronyms ‘NA’ and ‘NUC’ in the Figure legend for Figure 1

Answer:

The authors thank the Reviewer for the right observations and have followed the indications given. Moreover, the text has been revised by a native speaker of English

We thank the Reviewers for helping us to improve our paper.

The address for correspondence is: Nicola Coppola, Department of Public Medicine, Section of Infectious Disease, University of Campania, via L. Armanni 5, 80133, Naples, ITALY; Tel +39 081 5667718; Fax: +39 081 5667719; e-mail: nicola.coppola@unicampania.it.

We hope that the paper is now worthy of publication in VIRUSES

Naples, September 02 2019

Yours sincerely                                              

Nicola Coppola

Reviewer 2 Report

Manuscript ID: viruses-586419

Type of manuscript: Review

Title: HEPATITIS B VIRUS (HBV) REACTIVATION FOLLOWING PHARMACOLOGICAL 

ERADICATION OF HEPATITIS C VIRUS (HCV)

Authors: Marinatonietta Pisaturo, Margherita Macera, Loredana Alessio, 

Federica Calò, Nicola Coppola

Major;

There was no report supported management of HBV infection during and after DAA-based treatment in Figure 1. For example, in case of cirrhotic patients, treatment for both HBV/HCV infection should be continued, irrespective of HBV viral load. How about it?

Minor;

Page 7 of 13; The data reported by Macera M was incorrect. 5 (17%)/0

Page 9 of 13; authors should show the unit of HBV DNA in Figure 1.

There was wrong numbering of the referred papers between no. 51 to no. 61.

Author Response

Dear Editor,

We re-submit to your attention our paper Hepatitis B Virus (HBV) Reactivation Following Pharmacological Eradication of Hepatitis C Virus (HCV) ( Viruses-586419 ) modified according to the suggestions of the reviewers.

ANSWER TO THE COMMENTS OF REVIEWER 2

 Reviewer: 2

Comments to the Author  

Major;

There was no report supported management of HBV infection during and after DAA-based treatment in Figure 1. For example, in case of cirrhotic patients, treatment for both HBV/HCV infection should be continued, irrespective of HBV viral load. How about it?

Answer:

The authors thank the Reviewer for the observations and have added in the new manuscript (text and figure) the indications about the management of HBV overt/HCV cirrhosis.

Minor;

Page 7 of 13; The data reported by Macera M was incorrect. 5 (17%)/0

Answer:

The authors have corrected data as indicated.

Page 9 of 13; authors should show the unit of HBV DNA in Figure 1.

Answer:

The authors have indicated the unit of HBV DNA in Figure 1.

There was wrong numbering of the referred papers between no. 51 to no. 61.

Answer:

The authors have corrected the numbering as indicated.

We thank the Reviewers for helping us to improve our paper.

The address for correspondence is: Nicola Coppola, Department of Public Medicine, Section of Infectious Disease, University of Campania, via L. Armanni 5, 80133, Naples, ITALY; Tel +39 081 5667718; Fax: +39 081 5667719; e-mail: nicola.coppola@unicampania.it.

We hope that the paper is now worthy of publication in VIRUSES

Naples, September 02 2019

Yours sincerely                                              

Nicola Coppola

Round 2

Reviewer 1 Report

The manuscript is improved and the authors have addressed my concerns.